# Ductal Adenocarcinoma of the Prostate with Novel Genetic Alterations Characterized by Next-Generation Sequencing

**Alexandra Zara Rozalen [1,2], Jose Manuel Martin [1], Rithika Rajendran [1], Maneesh Jain [3,4] and Victor E. Nava [1,5,*]**

[1] Department of Pathology, Veterans Affairs Medical Center, Washington, DC 20422, USA; alexandra.zararozalen@mountsinai.org (A.Z.R.); martincastellimd@gmail.com (J.M.M.); rithika.rajendran@va.gov (R.R.)

[2] Department of Pathology, Mount Sinai Morningside and West Hospitals, New York, NY 10019, USA

[3] Section of Hematology & Oncology, Veterans Affairs Medical Center, Washington, DC 20422, USA; maneesh.jain@va.gov

[4] Department of Medicine, The George Washington University Hospital, Washington, DC 20037, USA

[5] Department of Pathology, The George Washington University Hospital, Washington, DC 20037, USA

\* Correspondence: victor.nava@va.gov; Tel.: +1-202-745-5749

**Abstract:** Ductal adenocarcinoma of the prostate (DAP) is an uncommon variant of prostate cancer associated with aggressive disease and poor outcome. It presents most frequently as a mixed tumor combined with acinar adenocarcinoma. Although the histopathological features of DAP are well known, its genomic characteristics are still evolving, prompting the suggestion that all DAP would benefit from molecular analysis with the purpose of improving tumor recognition, genetic classification, and, ultimately, personalized therapy. Herein, we report a case of DAP with novel genetic alterations (BCOR P1153S, ERG M219I, KDR A750E, POLE S1896P, and RAD21 T461del).

**Keywords:** ductal adenocarcinoma; prostate cancer; next-generation sequencing; genetic mutations

## 1. Introduction

Ductal adenocarcinoma of the prostate (DAP) is a rare subtype of prostatic carcinoma (PCa); it is composed of tall columnar cells arranged in a cribriform, papillary, or solid pattern that resembles the intestinal/endometrioid epithelium arising from the periurethral or peripheral prostatic ducts, which can be differentiated from high-grade prostatic intraepithelial neoplasia and intraductal carcinoma by the absence of basal cells [1,2]. DAP is the second most common malignant tumor of the prostate, after acinar adenocarcinoma (AAC), and commonly presents as a mixed tumor with the latter in Caucasian males older than 70 years of age [3]. The reported incidence of DAP varies and can be as high as 3 to 5% of all prostate cancers [4,5]. However, a recent meta-analysis revealed that pure DAP represents less than 1% of all PCa [6]. Although a consensus regarding the biology, diagnosis, treatment, and outcome of DAP is still lacking, most authorities regard this tumor as more aggressive than AAC, which correlates with a higher rate of extracapsular extension (T3) and metastatic disease [6]. Therefore, the comprehensive molecular characterization of DAP may be helpful in identifying actionable targets for precision management [7]. Herein, we present a case of DAP with novel genetic alterations characterized by next-generation sequencing (NGS).

## 2. Detailed Case Description

A 66-year-old black man presented with intermittent, painless gross hematuria and a weak urine stream. A previous clinical history of a laryngectomy for T3N0M0 squamous cell carcinoma of the larynx that was recurrent after chemotherapy and radiotherapy was noted. Prostate-specific antigen (PSA) was elevated (5.8 ng/mL), and a transrectal needle biopsy of the prostate revealed DAP (Gleason score 4 + 3 = 7, ISUP grade group 3) involving

11 of 12 cores (Figure 1). Cystoscopy detected three papillary urethral tumors (measuring 0.2 cm, 0.5 cm, and 1 cm, respectively), which could not be sampled due to poor visibility secondary to bleeding. Two months later, the urethral tumors were biopsied on a repeat cystoscopy, confirming metastatic DAP that was positive for NKX3.1, PSAP, and PSA and negative for P63 by immunohistochemistry. One of these tumors was considered to be a direct extension of the primary tumor (pathologic stage T4), involving the proximal urethra. However, a second tumor in the distal urethra was separate from the primary tumor and surrounded by normal tissue and was thus considered to be a metastasis (pathologic stage M1). NGS (FoundationOne CDx-324 gene panel) was performed only in the primary tumor, and it identified five pathogenic mutations (AKT1 E17K, BRAF-AGAP3 fusion, CTNNB1 splice site 242-1G>A, MLL2 W4377*, and TP53 Y220C), as well as six variants of uncertain significance (BCOR P1153S, CDK12 R902L, ERG M219I, KDR A750E, POLE S1896P, and RAD21 T461del). The tumor mutation burden (six per megabase) was low, and microsatellite instability was not detected. Imaging studies, including computed tomography, bone scans, and positron emission tomography did not reveal additional metastatic disease or adenopathy. Multiparametric magnetic resonance imaging of the prostate revealed a 3.1 × 3.0 cm eccentric T2 hyperintense lesion in the apex, which was inseparable from the prostatic urethra and the colon. However, extra-prostatic extension was not recognized. Despite a stable clinical course, hematuria persisted. After multidisciplinary evaluation, a surgical approach was not considered for this patient due to his age and comorbidities. He was treated with leuprolide acetate in combination with radiotherapy (urethral implant), resulting in the resolution of hematuria and the progressive decrease in PSA from 5.8 ng/mL to 0.37 ng/mL; the patient remained in remission 28 months after diagnosis.

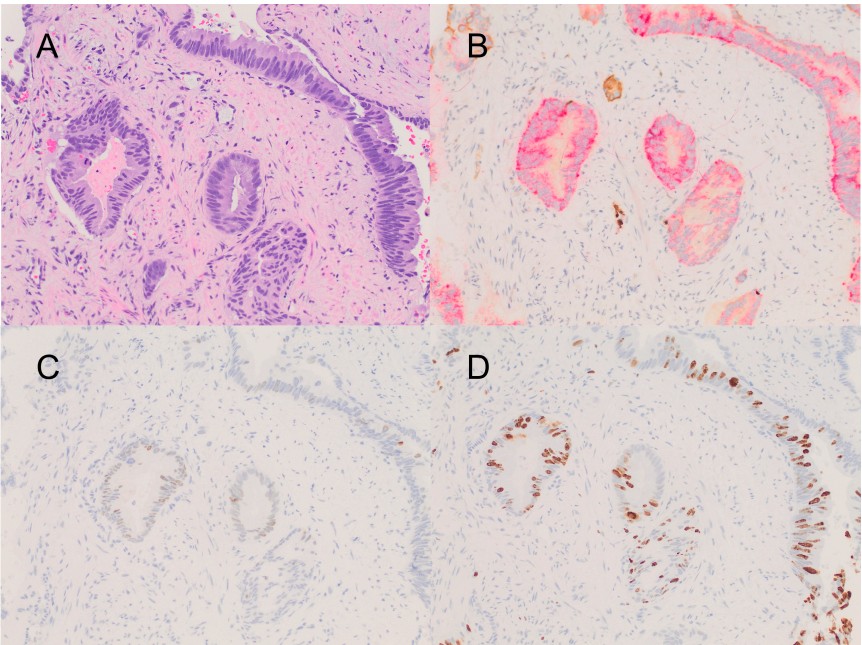

**Figure 1.** (**A**) Ductal adenocarcinoma composed of tall columnar cells with occasional mitoses (Hematoxylin and Eosin stain). (**B**) Immunohistochemistry for PIN4 triple stain demonstrating cytoplasmic positivity for racemase on tumor glands (red stain), which are devoid of basal cell layers (brown stain for P63 and keratin 34 beta E12). (**C**) Immunohistochemistry for P53 showing variable nuclear expression on tumor cells. (**D**) Immunohistochemistry for Ki-67 showing tumoral proliferative rate of ~40%. All at 200×.

## 3. Discussion

DAP is generally characterized by a suboptimal response to androgen deprivation, which may be improved by targeted therapies aimed at specific molecular alterations. However, information on the molecular underpinnings of DAP is limited. Although the genomic

overlap between DAP and AAC is considerable, unique differences in gene expression remain significant [8]. TMPRSS2:ERG fusions are less common in DAP [9], but mutations involving the PI3K-AKT and Wnt pathways are frequently shared between DAP and AAC. Phosphatase and tensin homolog (PTEN) alterations and CTNNB1 hotspot mutations are enriched and mutually exclusive in DAP. Interestingly, both histologic subtypes of PCa seem indistinguishable in terms of androgen receptor (AR) expression, but DAPs are typically less responsive to androgen deprivation therapy (ADT) than AAC [10]. Recent multi-institutional research identified at least 1 DNA damage repair (DDR) gene alteration in 49% of 51 cases of pure DAP or in the ductal component of mixed carcinomas, including germline pathogenic mutations in 20% [7]. Due to its poor prognosis, it has been suggested that all DAPs could benefit from the use of molecular analysis to identify actionable therapeutic targets and potentially improve outcomes. Accordingly, we performed NGS on one case of pure DAP, detecting well-known and novel alterations [11].

The well-known alterations include the activation of the mutations of the PI3K-AKT, MAPK (BRAF-AGAP3 fusion), and Wnt (CTNNB1 splice site 242-1G>A) pathways, MLL2 W4377* and CDK12 R902L. In addition, one inactivating mutation of p53 (TP53 Y220C) was identified.

The E17K substitution in AKT1/RAC (Rho family)-alpha serine/threonine-protein kinase was reported in 1.4% of all prostatic carcinomas (ductal and acinar) in a large series, and it activated AKT1 independently of PI3K. Notably, E17K is mutually exclusive with PTEN/PIK3CA mutations and is associated with a more favorable prognosis that is independent of the histologic pattern (acinar vs. ductal) [12]. Although B-RAF proto-oncogene (BRAF) rearrangements have been reported in up to 2.5% of PCa [13], BRAF fusions with AGAP3 (ARF-GAP with GTPase/ANK repeat and PH domain-containing protein 3/CENTG3) are rare. They activate the BRAF kinase domain, driving the MAPK pathway, and represent only 0.06% of AACR GENIE cases, including thyroid gland papillary carcinoma, colon adenocarcinoma, and low-grade glioma, but not prostatic primaries [14]. Incidentally, a BRAF-AGAP3 fusion was described in metastatic melanoma [15], and a related BRAF-NUT fusion was previously described in prostatic AAC [15]. The specific mutations found in β-catenin/CTNNB1, Mixed Lineage Leukemia 2 (MLL2)/Histone-lysine N-methyltransferase 2D (KMT2D/KABUK1/MLL4), and TP53 are well characterized in PCa [16]. Of note, many mutated forms of Cyclin-dependent kinase 12 (CDK12) have been detected in several cancers, including PCa, with a significantly higher rate in metastatic tumors [17]. However, R902L is not retrievable in PubMed, ClinVar, COSMIC, or the Human Gene Mutation Database in association with PCa.

Remarkably, five of the alterations (BCOR P1153S, ERG M219I, KDR A750E, POLE S1896P, and RAD21 T461del) have not yet been published to the best of our knowledge. Other BCL-6 co-repressor (BCOR) mutations were reported in high-grade sarcomas [18] and acute myeloid leukemia [19]. Interestingly, this gene regulates the proliferation of prostate cancer cell lines in an androgen-dependent manner [20], which may correlate with the observed therapeutic response. Mutations in transmembrane protease serine 2/PRSS10 (TMPRSS2) are common in PCa, and a fusion with the ETS-related gene/v-ets avian erythroblastosis virus E26 oncogene homolog (ERG) is seen in more than 30% of cases [21,22]. Importantly, the expression of the ERG transcription factor is associated with better outcomes in PCa, which correlates with its AR-dependent function [23]. Kinase Insert Domain Receptor (KDR)/Vascular endothelial growth factor 2 (VEGFR2)/CD309 is a membrane tyrosine kinase involved in pleiotropic signaling, which regulates the angiogenesis of multiple cancers and is altered in 1.5% of PCa [14,24]. Remarkably, KDR A750E maps to the extracellular immunoglobulin constant 2 protein domain involved in ligand recognition. The DNA polymerase epsilon catalytic subunit/POLE gene is essential in DNA replication and is frequently altered in endometrial carcinoma, melanoma, and gastric adenocarcinoma [25]. Notably, only one POLE mutation (V511L) has been reported in high grade PCa with neuroendocrine differentiation [26]. Nevertheless, various POLE alterations are documented in the cBioPortal for Cancer Genomics in 2.63% of PCa cases

(0.81% are point mutations, 0.4% amplifications, and 1.42% deletions). Of interest, all the mutations (including A465V, A1495T, V240M, L1220P, and D392Gfs*25) were found in AAC but not in DAP [27]. Finally, an alteration in a RAD cell cycle checkpoint protein that bind chromatin in response to DNA damage was found. The RAD21 homolog (also known as MGS, HR21, MCD1, NXP1, SCC1, CDLS4, hHR21, and HRAD21) is an essential evolutionarily conserved protein of the a-Kleisins family that repairs DNA double-strand breaks. As a central component of the multi-protein cohesin complex, RAD21 has at least 285 interactors; accordingly, it is involved in multiple key cellular processes related to cell survival, proliferation, and motility. Aberrant RAD21 function is well documented in several non-malignant (cohesinopathies) and malignant diseases [28]. Mutated RAD21 has been detected in 2.48% of all cancers and represents adenocarcinomas of the breast, lung, colon, and prostate [14]. Furthermore, RAD21 overexpression is associated with more aggressive PCa and is frequently observed in BRCA-mutated prostatic tumors [29]. In addition, RAD21 is commonly amplified in advanced androgen-resistant PCa [30], suggesting alternative genetic mechanisms in our patient.

Even though diverse alterations in AR are associated with the development of resistance to ADT [31], this gene is a wild-type in our patient, which correlates with the excellent clinical response to leuprolide acetate and radiotherapy. In addition, recent evidence suggests that DAP has a similar response to hormone deprivation but a higher rate of early metastasis in comparison with high-grade AAC [32], and aggressive management with radiation and ADT portends longer disease-free survival with this tumor, in contrast to previous reports [33]. Based on the genetic profile, there was no clear evidence to suggest peculiar radiosensitivity since the tumor was microsatellite-stable. In fact, based on the TP53 mutated status, this tumor could be more resistant to radiation. Interestingly, some of the novel mutations were found to affect the genes involved in DNA replication/repair (POLE and RAD21), and it is not inconceivable that they contributed to the excellent clinical response observed.

In summary, we present a case of DAP with novel genetic alterations characterized by NGS. Five pathogenic mutations (AKT1 E17K, BRAF-AGAP3 fusion, CTNNB1 splice site 242-1G>A, MLL2 W4377*, and TP53 Y220C), as well as six variants of uncertain significance (BCOR P1153S, CDK12 R902L, ERG M219I, KDR A750E, POLE S1896P, and RAD21 T461del) were identified. Although the oncologic management was not changed for this patient, we believe that routine NGS for rare tumors may be beneficial in the identification of future druggable alterations. Our results support the utility of the molecular analysis of DAP to further characterize and possibly inform the management of this type of aggressive tumor variant.

## 4. Conclusions

DAP is an aggressive rare variant of prostate cancer, which may benefit from routine analysis by NGS. Five pathogenic mutations (AKT1 E17K, BRAF-AGAP3 fusion, CTNNB1 splice site 242-1G>A, MLL2 W4377*, and TP53 Y220C) and six variants of uncertain significance (BCOR P1153S, CDK12 R902L, ERG M219I, KDR A750E, POLE S1896P, and RAD21 T461del) from one case of DAP are reported. To the best of our knowledge, five of the identified alterations (BCOR P1153S, ERG M219I, KDR A750E, POLE S1896P, and RAD21 T461del) have not previously been published, and they contribute to the molecular landscape of this aggressive tumor.

**Author Contributions:** A.Z.R. wrote the first draft and contributed with figure preparation and reference organization. R.R. and J.M.M. contributed with a critical literature search and editing of the manuscript. M.J. contributed with refinement of the study design, manuscript review, and clinical care. V.E.N. designed the work, performed histopathologic diagnosis, took micrographs, and conducted a critical review of the initial and final drafts. All authors have read and agreed to the published version of the manuscript.

**Funding:** This research received no external funding.

**Institutional Review Board Statement:** Ethical review and approval exempted by the Veterans Affairs Washington DC Medical Center Institutional Review Board for case reports.

**Informed Consent Statement:** Waiver of informed consent was obtained from the IRB according to institutional guidelines.

**Data Availability Statement:** Datasets may be publicly available upon request pending decision of the National Precision Oncology Program of the Veterans Affairs.

**Conflicts of Interest:** The authors declare no conflicts of interest.

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
