# Peer review of "Ductal Adenocarcinoma of the Prostate with Novel Genetic Alterations Characterized by Next-Generation Sequencing"

_curroncol, doi:10.3390/curroncol31030118_

Round 1

Reviewer 1 Report

Comments and Suggestions for Authors

Overall interesting case report and clearly reported, however a few key questions. 

1. What was the patient's overall status and disease at last follow-up? Currently the authors only elude to an excellent response to the described therapy. 

2. Was the patient a surgical candidate at any point in time? If so, please indicate and if not, please specify the contraindications or discussion regarding deferral of resection. 

3. Is there any data to suggest that any of the described gene alterations could contribute to radiosensitivity?  

4. This manuscript would benefit from additional discussion on how the genetic alterations influenced the authors decision on the specific cancer therapy. Would all patients with DPA benefit from genetic sequencing of their tumors?

Reviewer 2 Report

Comments and Suggestions for Authors

This is an interesting case report detailing the pathological and genomic features of a prostate cancer case with ductal adenocarcinoma of the prostate. There is some lack of clarity in the methods applied, making it difficult to determine the validity of the conclusions. I realise that the case report format makes it difficult to include methodological details, but you can presumably include these as supplementary materials..? I have the following comments:

- The authors say that two urethral tumours were biopsied (page 2, line 46), one primary and one metastatic - how was this determined? was the sequenced tumour from the primary or metastatic or both? If both, was there any difference in mutations between these two tumours?

- Some information on the NGS platform used would be helpful - how many genes and bases does this cover, what is the depth of sequencing, how was pathogenecity of the mutations determined (which database was used)

- The authors state that data sharing is not applicable in the Data Availability statement, but surely the NGS data (fastqs or aligned bams, vcfs) can be shared, perhaps on dbGAP or European Genome phenome Archive?

Round 2

Reviewer 2 Report

Comments and Suggestions for Authors

NA